# Addressing Weight Bias in the Cisgender Population: Differences between Sexual Orientations

**DOI:** 10.3390/nu14091735

**Published:** 2022-04-22

**Authors:** Paolo Meneguzzo, Enrico Collantoni, Valentina Meregalli, Angela Favaro, Elena Tenconi

**Affiliations:** 1Department of Neuroscience, University of Padova, 35121 Padova, Italy; enrico.collantoni@unipd.it (E.C.); valentina.meregalli@gmail.com (V.M.); angela.favaro@unipd.it (A.F.); elena.tenconi@unipd.it (E.T.); 2Padova Neuroscience Center, University of Padova, 35129 Padova, Italy

**Keywords:** weight bias, gay, lesbian, bisexual, obesity, cisgender, heterosexual

## Abstract

(1) Background: Weight bias (WB) is an implicit psychological construct that can influence attitudes, beliefs, body experience, and evaluation of specific psychopathology relationships. Sexual orientation has played a crucial role in developing and maintaining psychiatric conditions linked to body evaluation, but few studies have evaluated possible connected biases. Thus, the paper aims to assess potential relationships between sexual orientation and WB, looking at potential roles in specific psychopathology; (2) Methods: A total of 836 cisgender subjects participated in an online survey, aged between 18 and 42 years old. Two specific aspects of WB were evaluated with validated scales about beliefs about obese people and fat phobia. Demographic variables, as well as depression and eating concerns were evaluated; (3) Results: Gay men and bisexual women showed higher levels of fat phobia, depression, and eating concerns. Regression analysis showed that sexual orientation significantly predicted fat phobia (*p* < 0.001) and beliefs about obese people (*p* = 0.014); (4) Conclusions: This study confirms the vulnerability of gay men and bisexual women to cognitive bias about their own bodies, showing a potential vulnerability about body and weight concerns.

## 1. Introduction

Weight bias (WB) is defined as negative weight-related attitudes, beliefs, assumptions, and judgmental stereotypes that negatively influence body experience and evaluation [1]. This bias can be a crucial element in evaluating one’s own body, distorting real perceptions of body shape and weight, as well as reinforcing negative emotions with implicit biases that could influence one’s judgment [2,3]. The literature has shown that WB leads to negative emotions and concerns about one’s body and weight, and may correlate to unhealthy eating behaviors, low quality of life, depression, and maladaptive behaviors [4]. A recent “call to action” paper has shown the role of WB across the weight spectrum, encouraging an integrative perspective on obesity and eating disorders and the need to reduce the influence of weight standards in society in order to increase the quality of life and mental wellbeing [5]. The same has been seen in people with higher weight, where WB has been shown to be related to body weight misperceptions after bariatric surgery [6], with possible negative surgery outcomes. Indeed, WB is linked to adverse mental health outcomes and may play a crucial role in the development or maintenance of body image dissatisfaction and pathological eating behaviors [7], with a possible negative impact on the outcome of treatments [8].

Data have shown that WB differs with respect to individual characteristics such as age, sex, and body weight [9], suggesting different strategies for its reduction. Looking only to gender differences, women reported higher levels of WB, with a higher connection to eating disorders and depression [10,11]. Only a few studies about WB have considered sexual orientation, with various methodologies focusing mainly on differences between gay men and heterosexual peers [12,13]. Weight stigma is a result of WB, and it is defined as discriminatory acts and thoughts targeted towards individuals because of their weight [14]. Weight stigma is a common experience in sexual minority people, and it is linked to poorer quality of life and higher levels of internalization of weight bias ideas, with consequent psychological distress [1,14,15]. Even if the weight stigma phenomenon derives from cognitive bias and is equally harmful to all men despite their sexual orientation, gay and bisexual men displayed heightened WB levels compared to their heterosexual peers [13,16]. Weight stigma is something that gay and bisexual men reported both from others and for themselves [13], but few studies are available about their judgment style about themselves. Few studies are available on WB in sexual minority women, perhaps because body image and related cognitive features were for many years not considered a problem within this group [17]. New evidence has shown, however, that bisexual women (BIW) frequently experience adverse thoughts about their bodies [18] as well as misperceive their weight status with health-related consequences. Moreover, a growing body of literature has shown the presence of differences between BIW and lesbians regarding body image concerns, body misperception, and eating behaviors [17,18], indicating that WB should be investigated in this population as well. Moreover, sexual minority people who reported weight-based victimization also reported lower levels of quality of life and lower own-health perceived [19], showing the need for a deeper understanding of the possible role of sexual orientation in WB.

Converging evidence has shown that minorities’ stress due to sexual orientation could have a role in dysfunctional eating behaviors [18,20], but no specific conclusion about the relationship between these aspects has been drawn. Data are still preliminary, but recent studies have found both in sexual minority men and women the presence of higher BMI, higher levels of binge eating, and higher internalization of weight biases [21,22]. Thus, considering the role of WB in the quality of life and wellbeing [23], more studies are needed in the sexual minority population that could explain the presence and the relationship of WB with eating concerns and general wellbeing.

For all these reasons, our primary goal was to evaluate the role of WB across the sexual orientation spectrum, looking for a specific connection to eating concerns and correlated behaviors which we hypothesized to differ across specific sexual orientations. Moreover, we hypothesized that due to the disparities that emerged in WB between genders and sexual orientations, BIW and lesbians could be the subgroups with worse scores.

## 2. Materials and Methods

The participants were recruited via online invitations through social media (i.e., Italian Facebook groups related to gender, physical activities, and cultural associations linked to civil rights; both open and close groups) and LGBTQ+ group mailing lists from the area of the Veneto Region (Italy), through those responsible for managing personal data, without the involvement of researchers. The invitation consisted of a request to complete voluntary and spontaneous questionnaires on body image and body experiences and indicated that the questionnaires would be used for research purposes, as suggested by the previous literature [24]. The online survey was devised in such a way as to prevent multiple responses from the same IP addresses, but the IP addresses were hidden from investigators. The online survey did not allow multiple responses from the same IP address, and IP addresses were not linked to the answers. No participants received any remuneration for their participation.

The data, collected between September 2019 and March 2020, explore the role of specific cognitive aspects such as fat phobia, defined as a pathological fear of fatness, and beliefs about obesity in the cisgender population, looking for these constructs’ function in the psychopathology linked to eating behaviors and thoughts, as well as their relationships to different sexual orientations. The inclusion criteria were: (1) written informed consent obtained before the questionnaires; (2) ≥18 years of age; (3) fluent understanding of written Italian. No specific exclusion criteria were applied.

Each participant provided written informed consent agreeing to participate in the survey. The research was in accordance with the Declaration of Helsinki, its later amendments, and local legislation about anonymous questionnaires, according to the local Ethics Committee.

### 2.1. Measures

We asked the participants to provide demographic information such as age, race, education, height, and weight. The body mass index (BMI) was calculated as weight in kilograms divided by height in meters squared, using the data given by participants. The gender of participants was determined by specific items asking them to self-identify as “cisgender,” “transgender,” or “non-binary”. The sexual orientation was self-identified by each participant as “heterosexual,” “bisexual,” “gay/lesbian,” or “asexual”, as previously applied [18]. Other psychological constructs that were evaluated according to the previous literature data and the aim of the study were depression and eating concerns as general psychological factors correlated with WB, and fat phobia and beliefs about obese people as specific elements of WB.

The Patient Health Questionnaire-9 (PHQ-9) is a screening tool for depression, with robust evidence of sensibility and sensitivity [25]. It is a nine-item scale, where each item evaluates the presence of one of the DSM criteria for a depressive episode in the prior two weeks. Answers were forced into a Likert scale with four possible choices: 0 (“not at all”), 1 (“a few days”), 2 (“more than half the days”), and 3 (“almost every day”), with higher scores indicating higher depressive symptomatology. In this study, Cronbach’s α = 0.79.

The Eating Attitudes Test (EAT-26) is a widely used self-reported questionnaire that collects information about symptomatology and eating disorder-related concerns [26]. It is composed of 26 items, rated on a six-point Likert scale ranging from 1 (“never”) to 6 (“always”). Higher scores indicate higher eating concerns, with a clinical cut-off score at 20 points. In this study, Cronbach’s α = 0.91.

The Beliefs About Obese Persons Scale (BAOP) is a validated, self-administered eight-item questionnaire used to evaluate beliefs about the causes of obesity [27]. Higher scores imply a stronger belief that obesity is not under the person’s control, indicating a lack of negative judgment of people at higher body weight and less WB. In this study, Cronbach’s α = 0.75.

The Fat Phobia Scale (FPS) is a 14-item questionnaire measuring negative attitudes toward higher weight [28]. The participants are asked to imagine a specific person characterized by a high weight, and they must indicate on a scale from 1 to 5 which adjective they feel best describes that person’s feelings and beliefs (e.g., “no will power” versus “will power”), showing the degree of their stereotypical assumptions about being fat. Scores lower than 2.5 denote negative attitudes. In this study, Cronbach’s α = 0.77.

### 2.2. Statistical Analysis

An a-priori power analysis for a 2 (gender)  ×  3 (sexual orientation) MANOVA was conducted using G*Power vers. 3.1.9.7 (Universität Düsseldorf, Düsseldorf, Germany) [29], assuming f^2^  =  0.0625, α = 0.05, 1-β = 0.95, indicating that the total sample size should be 213. The variance analyses were performed with the Kruskal–Wallis test, due to the non-parametric nature of most of the study’s data. Post hoc comparisons were performed using Bonferroni correction. Correlation analyses were performed by Spearman’s ρ. Regression analyses were conducted with sexual orientation as an independent variable using heterosexual subgroup results as dummy variables and setting FPS and BAOP as dependent variables. The alpha was set at *p* < 0.05 for all of the analyses, and the effect sizes were calculated with partial eta squared. Bonferroni corrections for multiple testing have been applied by dividing 0.05 by the overall number (4) of questionnaire comparisons, with the level of significance set at 0.013. The entire analysis was conducted with IBM SPSS Statistics 25.0 (SPSS, Chicago, IL, USA).

## 3. Results

A total sample of 940 people decided to open the online survey. We excluded all the incomplete responders and all the responders faster than 5 min to exclude bot responders (*n* = 90, 9.6% of the responders). We also excluded all the responders who identified themselves as transgender or non-binary (*n* = 12), due to the low number of responders. Only eight women identified themselves as asexual, and for statistical reasons, were excluded, while no men identified themselves as asexual. We obtained a total sample of 830 cisgender individuals.

The sample was then composed of 506 women (53.8% of the participants). Most of the sample described themselves as white (98.3%) and engaged a relationship of any kind (78.5% of the participants). Due to the nature of the questionnaire, the totality of the participants were Italian speakers. Please see Table 1 for the demographic details of the participants.

No significant differences emerged between subgroups regarding age (F (824,5) = 1.720, *p* = 0.127), but BMI showed a significant difference between subgroups (F (824,5) = 2.406, *p* = 0.035), with heterosexual women (HEW) having a lower BMI than bisexual men (BIM, *p* = 0.003) and heterosexual men (HEM, *p* < 0.001) at the post-hoc analysis with Bonferroni correction. The psychometric assessment showed significant differences in specific psychological domains across the sexual orientation spectrum, as shown in Table 2. Graphical representation of the FPS scores showed the different distribution of the results across the sexual orientation spectrum (see Figure 1).

Several correlation analyses were performed seeking relationships between the included constructs. Table 3 reports the results of the correlations.

Regression analysis was performed for both of the constructs linked to WB included in this survey, looking for significant relationships between constructs and demographic data and considering genders separately. In women, we found no significant regression for both FPS and BAOP using PHQ-9, EAT26 TOT, BMI and age as predictors. In the male subsample, we found that both FPS and BAOP were significantly predicted by EAT-26 TOT alone [FPS: R^2^ = 0.17, F (1,322) = 9.90, *p* = 0.002; BAOP: R^2^ = 0.21, F (1,322) = 14.72, *p* < 0.001] and also with BMI [FPS: R^2^ = 0.22, F (1,321) = 8.41, *p* < 0.001; BAOP: R^2^ = 0.17, F (1,322) = 10.92, *p* < 0.001]. Regarding the role of sexual orientations, different regressions were performed using heterosexual scores as comparison using dummy variables, showing different regression coefficients in sexual minority groups, see Table 4 for details.

## 4. Discussion

The primary goal of the paper was to evaluate the presence of different levels of weight bias (WB) across the sexual orientation spectrum in both genders. In particular, we were interested in data about BIW and reinforcing previous evidence about gay men’s body and weight concerns.

Our analyses showed that gay men had the highest levels of fat phobia, a particular aspect of WB, and the lowest scores on the BAOP scale, which indicates higher levels of negative beliefs and more blaming of people at higher body weights. The same occurred in women, where sexual minority individuals showed higher weight biases than heterosexual women. To the best of our knowledge, this is the first time that WB has been investigated across different sexual orientations in women. Our investigation focused on the cognitive aspects of the evaluation of a person’s body weight. Data in the literature have shown that sexual minority men perceive higher levels of weight stigma [13,30], and the same has been found for lesbians [31]; however, to our knowledge, no studies have investigated the role of WB. Weight bias is a construct that may play a role in the management of body image and could be linked to unhealthy eating behaviors [6,7,32].

Moreover, WB has a role in poor interpersonal relationships and quality of life and could be linked to the physical health impairment of the individuals [23]. Recently, however, the literature has shown that it is possible to modify implicit and explicit biases with specific WB modification interventions [5], thereby pointing to new clinical applications. Our data confirmed the previously shown relationship between gay men and body shape and weight concerns, calling for specific assessments regarding body image in gay patients [33]. Indeed, bisexual women and gay men appeared as the subgroups with the highest level of weight stigma. This aspect is likely due to stress resulting from social stigmatization, which plays a significant role in developing an eating disorder in sexual minorities [34]. This should be considered when devising campaigns for body confidence to improve the wellbeing of gay men and BIW and reduce their body dissatisfaction [35].

Another notable result showed that a similar approach to body weight in BIW and gay men is the absence of a positive correlation between age and BMI present in all other sexual orientation groups. This data may corroborate the idea that there is an active control over one’s own body weight in BIW and gay men, showing the effects of a cognitive vulnerability to specific body shapes and weights that maybe be driven by WB [1,36]. Indeed, BIW and gay men are the two groups with the highest body weight dissatisfaction across the sexual orientation spectrum, and this concern could lead to weight control behaviors [18].

Depression and eating concerns have been pointed out as risk factors for body image disturbances and WB [37]. Our data showed the highest score for both these psychological domains in bisexual and lesbian women, corroborating the previous findings of the impaired psychological wellbeing in sexual minority women [38]. Moreover, we found significant relationships between fat phobia and depression scores in bisexual women, showing that this could be a vulnerability aspect that could explain previous results about body image evaluation [39] and therefore should be deeply evaluated.

There are some limitations to this study that need to be considered and which could be a starting point for necessary future research. Firstly, this study’s sample is from an online survey, and we must be careful not to overgeneralize the results to the whole population. Future studies could employ a double recruitment channel, using a statistical approach to limit sample size differences. Moreover, no robust data are available about the distribution of sexual minority population over the general population, limiting the generalization of our results. Secondly, we did not consider all of the possible constructs that the literature has shown to be potentially important aspects of socio-cultural pressure on sexual minority groups, such as internalized stigma, stress minority evaluation, and lack of social support. These should be investigated in future studies to examine their role on levels of weight stigma.

## 5. Conclusions

Despite the limits of this study, our WB evaluation across the sexual orientation spectrum has shown convergent evidence about the role of body weight and shape concerns in gay men and bisexual women. This study confirms the vulnerability of gay men and bisexual women in cognitive bias about their own bodies, showing a potential vulnerability about body and weight concerns. Due to the implicit role of body judgment, WB should be taken under serious consideration in treating body image concerns according to the sexual orientation of the clients. More studies concerning the relationship between WB and other cultural constructs are needed to achieve a better understanding of its function in the interpersonal domain, psychopathological constructs, and mental health.

## Figures and Tables

**Figure 1 nutrients-14-01735-f001:**
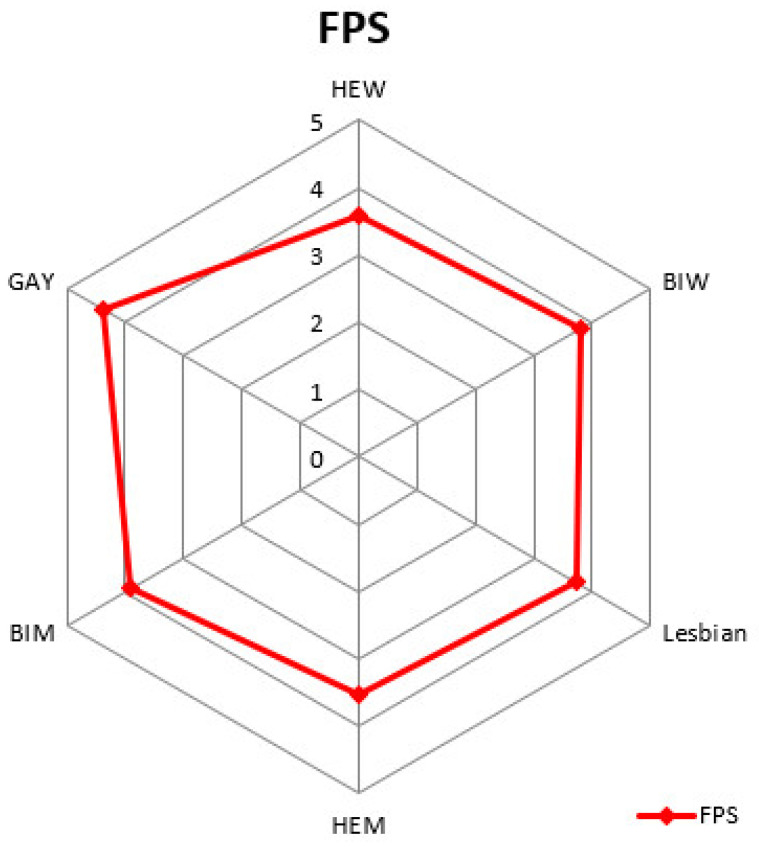
The figure shows the FPS results divided by sexual orientations. It is possible to appreciate how differently the results are distributed in the sexual minority groups of both genders compared to heterosexual peers. Gay men showed the highest scores for fat phobia in all groups included in the study, see Table 2 for data.

**Table 1 nutrients-14-01735-t001:** Demographic description of the participants.

		HEW *n* = 382	BIW *n* = 74	Lesbian *n* = 50	HEM *n* = 236	BIM *n* = 40	Gay *n* = 48
Age		26.69 (4.89) (18–40)	25.47 (3.94) (18–42)	26.42 (5.79) (18–38)	27.30 (5.50) (18–40)	26.43 (5.58) (18–35)	27.46 (5.49) (18–40)
BMI		22.56 (7.30) (15.57–58.60)	24.29 (6.10) (16.56–60.55)	24.92 (6.88) (16.90–41.50)	23.33 (3.53) (15.23–36.93)	22.48 (2.09) (20.01–28.21)	22.52 (3.40) (18.50–32.12)
Education	Lower secondary	2.3%	2.0%	7.0%	5.4%	2.0%	2.5%
	Upper secondary	25.9%	49.0%	39.5%	35.4%	40.5%	39.6%
	Degree	31.5%	30.6%	20.9%	27.7%	20.0%	11.8%
	Master or Doctorate	40.2%	18.4%	32.6%	31.5%	37.5%	46.1%
Relationship	Yes	76.4%	75.5%	79.1%	80.8%	77.9%	74.8%
	No	23.6%	24.5%	20.9%	19.2%	22.1%	25.2%

Means and standard deviations are reported, with minimum and maximum scores between brackets. HEW: heterosexual women; BIW: bisexual women; HEM: heterosexual men; BIM: bisexual men; BMI: body mass index, kg/m^2^.

**Table 2 nutrients-14-01735-t002:** Psychological evaluation of the sample.

	HEW *n* = 382	BIW *n* = 74	Lesbian *n* = 50	HEM *n* = 236	BIM *n* = 40	Gay *n* = 48	H	p η^2^_p_	Post-Hoc
PHQ9	8.77 (4.84)	11.13 (5.02)	9.70 (4.75)	6.98 (4.49)	7.65 (4.55)	6.85 (3.90)	56.797	<0.001 0.067	BIW > HEW (*p* = 0.001) HEW > HEM (*p* < 0.001) BIW > HEM (*p* < 0.001) BIW > BIM (*p* = 0.002) BIW > Gay (*p* < 0.001)
EAT26 tot	9.14 (10.17)	12.74 (13.19)	10.94 (9.25)	5.26 (4.57)	4.22 (3.67)	7.65 (4.57)	73.780	<0.001 0.073	HEW > HEM (*p* < 0.001) HEM > BIM (*p* = 0.011) BIW > HEM (*p* < 0.001) BIW > BIM (*p* < 0.001) Lesbian > HEM (*p* < 0.001)
FPS	3.57 (0.47)	3.82 (0.39)	3.73 (0.36)	3.52 (0.60)	3.90 (0.55)	4.36 (0.32)	112.111	<0.001 0.144	BIW > HEW (*p* = 0.003) BIW > HEM (*p* < 0.001) BIM > HEW (*p* = 0.002) Gay > HEW (*p* < 0.001) Gay > BIW (*p* < 0.001) Gay > Lesbian (*p* < 0.001) Gay > HEM (*p* < 0.001) Gay > BIM (*p* < 0.001)
BAOP	19.72 (3.73)	20.68 (3.64)	20.42 (4.08)	20.37 (4.46)	20.77 (1.94)	18.69 (2.36)	22.251	0.014 0.017	Gay < HEM (*p* = 0.004)

HEW: heterosexual women; BIW: bisexual women; HEM: heterosexual men; BIM: bisexual men; PHQ9: physical health questionnaire; EAT: eating attitude test; FPS: fat phobia scale, BAOP: beliefs about obese person. H: Kruskal–Wallis test for the evaluation of the distribution of variables with Pairwise Comparisons with Bonferroni correction.

**Table 3 nutrients-14-01735-t003:** Correlation analyses in different sexual orientation subgroups.

	Women															
		HEW					BIW					Lesbian				
		1	2	3	4	5	1	2	3	4	5	1	2	3	4	5
1	Age	-					-					-				
2	BMI	0.11	-				0.04	-				0.672 **	-			
3	PHQ 9	−0.19 **	0.08	-			0.01	0.03	-			−0.112	0.216	-		
4	EAT 26 tot	−0.10	0.08	0.48 **	-		0.10	0.15	0.43 **	-		0.191	0.160	0.33	-	
5	FPS	0.07	−0.07	−0.04	−0.08	-	−0.04	−0.05	−0.35 **	−0.08	-	−0.005	−0.086	0.10	−0.25	-
6	BAOP	0.06	−0.08	−0.04	−0.09	0.91 **	0.15	−0.05	−0.43 **	−0.14	0.83 **	−0.002	0.048	0.24	−0.18	0.68 **
	Men															
		HEM					BIM					Gay				
		1	2	3	4	5	1	2	3	4	5	1	2	3	4	5
1	Age	-					-					-				
2	BMI	0.26 **	-				0.55 **	-				0.05	-			
3	PHQ 9	−0.01	−0.07	-			−0.37	0.13	-			−0.37	0.06	-		
4	EAT 26 tot	−0.03	0.22 **	0.24 **	-		−0.65 **	−0.34	0.61 **	-		−0.28	0.09	0.55 **	-	
5	FPS	0.08	−0.03	0.07	0.25 **	-	0.01	0.01	0.01	−0.01	-	−0.32	0.07	−0.02	0.03	-
6	BAOP	−0.21 **	−0.01	−0.03	−0.22 **	−0.93 **	0.24	0.52 **	0.26	−0.46 **	0.09	0.19	0.07	−0.29	−0.51 **	−0.21

HEW: heterosexual women; BIW: bisexual women; HEM: heterosexual men; BIM: bisexual men; PHQ9: physical health questionnaire; EAT: eating attitude test; FPS: fat phobia scale, BAOP: beliefs about obese person. Spearman’s ρ is reported for each pair of variables. The significances are reported as = **: *p* < 0.01.

**Table 4 nutrients-14-01735-t004:** Regression analysis.

			Unstandardized Coefficients		Standardized Coefficient		
FPS	R^2^	*p*	B	SE	*β*	t	*p*
	0.039	<0.001	3.577	0.023		154.380	<0.001
BIW compared to HEW			0.238	0.058	0.183	4.144	<0.001
Lesbian compared to HEW			0.156	0.068	0.101	2.292	0.022
	0.224	<0.001	3.528	0.036		96.681	<0.001
BIM compared to HEM			0.370	0.096	0.192	3.859	<0.001
Gay compared to HEM			0.827	0.089	0.464	9.315	<0.001
BAOP							
	0.010	0.085	19.720	0.192		102.634	<0.001
BIW compared to HEW			0.956	0.477	0.090	2.004	0.046
Lesbian compared to HEW			0.700	0.565	0.056	1.240	0.216
	0.025	0.017	20.369	0.259		78.733	<0.001
BIM compared to HEM			0.406	0.680	0.033	0.598	0.550
Gay compared to HEM			−1.681	0.629	−0.149	−2.672	0.008

Regression analysis with sexual orientation as an independent variable. FPS: fat phobia scale, BAOP: beliefs about obese person.

## Data Availability

The datasets generated and analyzed during the current study are available from the corresponding author on reasonable request due to the institutional regulations.

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
