# Peer review of "Addressing Weight Bias in the Cisgender Population: Differences between Sexual Orientations"

_nutrients, 2022, doi:10.3390/nu14091735_

Round 1

Reviewer 1 Report

The article is of high quality and has been written correctly and with great precision in the details. Its reading has been simple and interesting, I think it has a great scientific quality. To increase the quality I make some suggestions:

- In the summary it would be good to add the age range of the sample, as in the methodology-participants section.

-In table 1 the term BMI appears. You must specify how you have measured it in the measurements section, if you calculated it yourself, if the participants calculated it themselves,... This is important information.

- Specify the inclusion and exclusion criteria since they are not specified as such and may cause problems in the generalization of the results.

- Add in each of the measures used its psychometric properties. Is Cronbach's α from your sample or the one found by the authors of each measurement? If it is not from your sample, you should calculate it.

Congratulations for the good job.

Author Response

Reviewer #1

The article is of high quality and has been written correctly and with great precision in the details. Its reading has been simple and interesting, I think it has a great scientific quality. To increase the quality I make some suggestions:

R: we thank Reviewer #1 for their supportive words.

- In the summary it would be good to add the age range of the sample, as in the methodology-participants section.

R: we appreciated this suggestion and included this information.

-In table 1 the term BMI appears. You must specify how you have measured it in the measurements section, if you calculated it yourself, if the participants calculated it themselves,... This is important information.

R: the BMI was calculated by authors using the height and the weight given by the participants. We explain this aspect in the methods.

- Specify the inclusion and exclusion criteria since they are not specified as such and may cause problems in the generalization of the results.

R: we agreed. We have now included exclusion and inclusion criteria.

- Add in each of the measures used its psychometric properties. Is Cronbach's α from your sample or the one found by the authors of each measurement? If it is not from your sample, you should calculate it.

R: we have now specified that the alphas are from this specific study.

Congratulations for the good job.

R: Thank you again!

Reviewer 2 Report

 The paper  is basically well constructed it just needs some fundamental methodological clarifications: two critical issues to be clarified: the representativeness of the sample with respect to which population (see below) and the post hoc statistical analysis method to avoid bias.

row 44 46 the cited paper binary divides the subjects therefore  is more appropriate to use "sex" rather than "gender"

row 62 a comment on lesbian versus bisexual may better clarify why choose bisexual as target

row 65 row 65 the data are similar, lower or higher than the remaining  of the population?

The distribution between groups (HEW, BIW; Lesbian, HEM, BIM, Gay) is it similar, different than the distribution in the national population? This is important given the potentially biased sample typology and should be addressed

Given the doubts about the objectivity of the "post hoc" analysis in statistics, it is necessary to insert a description of the used criteria to minimize bias conducting this analysis.

Author Response

Reviewer #2

 The paper  is basically well constructed it just needs some fundamental methodological clarifications: two critical issues to be clarified: the representativeness of the sample with respect to which population (see below) and the post hoc statistical analysis method to avoid bias.

R: we thank Reviewer #2 for the supportive comments and for the suggestions.

row 44 46 the cited paper binary divides the subjects therefore  is more appropriate to use "sex" rather than "gender"

R: we agreed and change the word.

row 62 a comment on lesbian versus bisexual may better clarify why choose bisexual as target

R: we included the evidence about the differences between bisexual women and lesbians as regards body concerns and eating behaviors.

row 65 row 65 the data are similar, lower or higher than the remaining  of the population?

R: lower. We have now included the direction of the comparison.

The distribution between groups (HEW, BIW; Lesbian, HEM, BIM, Gay) is it similar, different than the distribution in the national population? This is important given the potentially biased sample typology and should be addressed
R: Unfortunately, there is no robust data about the distribution of sexual minorities in the general population. We included this aspect in the limits of the paper, even if our approach is similar to other papers published in the literature about the representativeness of the sample. Moreover, we applied the most conservative approach for the analysis, using non-parametric analysis and considering a p-values correction for error type 1 - multiple analyses.  

Given the doubts about the objectivity of the "post hoc" analysis in statistics, it is necessary to insert a description of the used criteria to minimize bias conducting this analysis.

R: we included the description of the post hoc analysis applied. We used the pairwise comparisons with Bonferroni corrections, which is one of the most widely used one.